# Plasma Immune Proteins and Circulating Tumor DNA Predict the Clinical Outcome for Non-Small-Cell Lung Cancer Treated with an Immune Checkpoint Inhibitor

**DOI:** 10.3390/cancers15235628

**Published:** 2023-11-29

**Authors:** Simone Stensgaard, Astrid Thomsen, Sofie Helstrup, Peter Meldgaard, Boe S. Sorensen

**Affiliations:** 1Department of Clinical Biochemistry, Aarhus University Hospital, 8200 Aarhus, Denmark; simone.stensgaard@clin.au.dk (S.S.); sofiehelstrup@gmail.com (S.H.); 2Department of Clinical Medicine, Aarhus University, 8000 Aarhus, Denmark; petemeld@rm.dk; 3Department of Oncology, Aarhus University Hospital, 8200 Aarhus, Denmark

**Keywords:** non-small-cell lung cancer, immune checkpoint inhibitor, immunotherapy, immuno-oncology, biomarkers, circulating tumor DNA

## Abstract

**Simple Summary:**

A high PD-L1 expression in a tumor tissue biopsy qualifies lung cancer patients to treatment with an immune checkpoint inhibitor. However, tumor tissue is highly heterogeneous, and many patients progress early, despite having qualified for treatment. Thus, reliable biomarkers are lacking. Using blood samples, we aim to identify predictive biomarkers for responses to immune checkpoint-inhibitor treatment. Immune-related plasma proteins in lung cancer patients were quantified, and Fas ligand (FASLG) and inducible T-cell co-stimulator ligand (ICOSLG) were demonstrated to be associated with response and survival. In addition, combining these results with the quantity of circulating tumor DNA enabled us to identify a patient subgroup with prolonged survival. Liquid biopsies are a minimally invasive test, and our results demonstrate that they hold a predictive value supplementing the current eligibility criteria for treatment.

**Abstract:**

Immunotherapy has altered the therapeutic landscape for patients with non-small-cell lung cancer (NSCLC). The immune checkpoint inhibitor pembrolizumab targets the PD-1/PD-L1 signaling axis and produces durable clinical responses, but reliable biomarkers are lacking. Using 115 plasma samples from 42 pembrolizumab-treated patients with NSCLC, we were able to identify predictive biomarkers. In the plasma samples, we quantified the level of 92 proteins using the Olink proximity extension assay and circulating tumor DNA (ctDNA) using targeted next-generation sequencing. Patients with an above-median progression-free survival (PFS) had significantly higher expressions of Fas ligand (FASLG) and inducible T-cell co-stimulator ligand (ICOSLG) at baseline than patients with a PFS below the median. A Kaplan–Meier analysis demonstrated that high levels of FASLG and ICOSLG were predictive of longer PFS and overall survival (OS) (PFS: 10.83 vs. 4.49 months, OS: 27.13 vs. 18.0 months). Furthermore, we identified a subgroup with high expressions of FASLG and ICOSLG who also had no detectable ctDNA mutations after treatment initiation. This subgroup had significantly longer PFS and OS rates compared to the rest of the patients (PFS: 25.71 vs. 4.52 months, OS: 34.62 vs. 18.0 months). These findings suggest that the expressions of FASLG and ICOSLG at baseline and the absence of ctDNA mutations after the start of treatment have the potential to predict clinical outcomes.

## 1. Introduction

The therapeutic landscape for non-small-cell lung cancer (NSCLC) has advanced in recent decades with the developments of targeted therapy and immunotherapy [1,2,3]. Immunotherapy targeting the PD-1/PD-L1 axis has become part of the standard-of-care treatment of advanced NSCLC [4]. Eligibility for treatment with anti-PD-1/PD-L1 drugs is based on PD-L1 expression in a tissue biopsy measured as the tumor proportion score (TPS) by immunohistochemistry [5].

Pembrolizumab, an anti-PD-1 antibody, showed efficacy in the KEYNOTE-024 trial with prolonged progression-free survival (PFS) and overall survival (OS) in patients with NSCLC with a PD-L1 TPS of at least 50% as compared to standard platinum-based chemotherapy [6,7]. Additionally, an effect was observed in patients with a PD-L1 TPS as low as 1% in the KEYNOTE-042 trial [8]. PD-L1 TPS is, to a certain degree, associated with the response rate; unselected patients with NSCLC in the KEYNOTE-001 trial showed a response rate of 19.4% [9], while patients with NSCLC with a PD-L1 TPS of at least 50% in the KEYNOTE-024 trial had a response rate of 44.8% [6]. The OAK study investigating the efficacy of atezolizumab (an anti-PD-L1 antibody) showed that the treatment of PD-L1-negative NSCLC with atezolizumab led to improved survival rates compared to docetaxel treatment [10]. This underscores the suboptimal predictive value of PD-L1 expression, which, among other things, can be explained by a high degree of tissue heterogeneity [11,12,13]. In light of these findings, it becomes evident that additional biomarkers are needed.

Earlier studies have shown that immune cell infiltration in tumors, immune activation, and the pre-existing level of immunity are critical for the response to immunotherapy [14,15,16]. Since a solid tissue biopsy is highly invasive and may not represent the heterogeneity of the tumor, liquid biopsies in the form of blood samples are promising supplements to a solid tissue biopsy [17]. Blood samples enable the detection of plasma proteins and circulating tumor DNA (ctDNA) among others. These components can be used to detect disease, identify and monitor biomarkers for response or resistance, and predict progression [17,18,19,20,21,22,23]. For treatment response to immunotherapy, biomarkers for immune activation and activity are vital, and liquid biopsies may offer a minimally invasive way to detect and measure these.

In this study, we investigate immune-related plasma proteins and ctDNA in order to identify predictive biomarkers that can predict clinical outcome and survival rates.

## 2. Materials and Methods

### 2.1. Patients

Patients with advanced NSCLC treated with first- or second-line pembrolizumab at Aarhus University Hospital were eligible for enrollment. Treatment with pembrolizumab was initiated based on the PD-L1 TPS in the tumor tissue. The pathological detection of mutations in tissue biopsies was performed as part of the standard procedure for lung cancer patients at Aarhus University Hospital. The inclusion of patients occurred from January 2017 to April 2019. The patients received 2 mg/kg of pembrolizumab every 3rd week, and blood samples were collected before and during treatment. A total of 42 patients were included in this retrospective study from a previously described cohort [24]. All patients provided written consent. The study was approved by the regional ethics committee of Region Midt (approval number: 1-16-02-211-16) and was conducted in accordance with the Declaration of Helsinki.

### 2.2. Sample Collection and Preparation

Blood samples were collected before pembrolizumab treatment initiation and every third week afterward. The samples were collected in ethylenediaminetetraacetic acid (EDTA) tubes and centrifuged at 1400 g for 15 minutes, and the plasma was isolated and frozen at −80 °C. Plasma samples collected before treatment initiation (T_0_) and after one (T_1_) and two cycles of treatment (T_2_) were chosen for analysis. All patients had a T_0_ baseline sample and at least one sample from either T_1_ or T_2_ available. The T_0_ samples were collected 0–27 days before the start of pembrolizumab treatment (median: 2 days before). The T_1_ samples were collected after 17–43 days of treatment (median: 21 days), and the T_2_ samples were collected after 34–90 days of treatment (median: 43.5 days).

### 2.3. Olink Proximity Extension Assay

The levels of plasma proteins were analyzed by the Olink proximity extension assay (PEA) (Olink Proteomics, Uppsala, Sweden). The analysis was performed at BioXpedia, Aarhus, Denmark, following the manufacturer’s protocol. The Olink PEA technique is a qPCR-based assay, which allows for the detection and quantification of soluble proteins using a panel of antibodies. The panel contains an antibody pair for each protein of interest. The antibodies are conjugated to oligonucleotides, which are complementary for each pair of antibodies. The binding of the antibodies to the proteins brings the complementary oligonucleotides to a close proximity, and allows for the hybridization into a double-stranded molecule, which can be quantified using qPCR. The use of matched antibody pairs ensures that no cross-reactivity occurs, since only double-stranded oligonucleotides can produce an amplifiable qPCR product. A total of 1 µL of plasma was used for each sample and was mixed with the Olink Target 96 Immuno-Oncology panel (Olink Proteomics), which enabled the detection of 92 plasma proteins related to immuno-oncology. The samples were run on a 96-well plate along with a sample control, a negative control, and a plate control. A list of all proteins included in the panel can be found in Appendix A. The data are reported as normalized protein expression (NPX) values, which are relative quantification units that are logarithmically related to the protein concentration. The NPX was calculated from Ct values as follows: NPX_Sample_ = Correction Factor − (Ct_Sample_ − Ct_Extension Control_) − (Ct_Interplate Control_ − Ct_Extension Control_). The correction factor was a pre-determined factor that ensured that the background levels were zero. The correction factor was furthermore used to invert the values, so that a higher NPX value corresponded to a higher signal. Throughout this paper, the term protein expression refers to the NPX values.

### 2.4. Next-Generation Sequencing

The targeted next-generation sequencing of ctDNA was performed as part of a previously published study on the same patient cohort [24].

### 2.5. Statistical Analysis

The PFS was defined as the time from treatment initiation to the date of detection of radiological progression, based on the response evaluation criteria in solid tumors (RECIST) v. 1.1, or death due to any cause. The patients who had not progressed at the data cutoff date (20 December 2021) were censored at the date of their last radiological evaluation. OS was defined as the time from treatment initiation to death due to any cause, and patients with incomplete survival data were censored at the data cutoff date. The best overall response was defined as the most favorable radiologically determined response in the treatment period.

Patients were stratified based on the median PFS to evaluate differences in protein expression. Differential NPX values between the groups were evaluated using a t-test or a Wilcoxon rank sum test depending on the normal distribution of the data, which were assessed with the Shapiro–Wilk’s test. The fold change in expression was calculated on a linear scale for each protein as the geometric mean of the protein in patients with an above-median PFS divided by the geometric mean of the protein in patients with a below-median PFS. Due to the exploratory nature of this biomarker discovery study, we determined that corrections for multiple testing would be a too conservative approach and could result in the exclusion of potential predictive biomarkers, and we, therefore, presented the unadjusted *p*-values. Proteins were excluded from analyses if the median NPX values in both groups were below the limit of detection. Hierarchical clustering on scaled data was performed using the pheatmap R package.

The ctDNA sequencing data were analyzed using AVENIO Oncology Analysis Software v. 2.0.0 (Roche). A sample was considered ctDNA mutation positive if any lung cancer-related mutations could be detected in the sequencing data following the filtering described in our previously published study [24]. If no mutations were detected, the sample was defined as ctDNA mutation negative.

Survival analyses were performed using Kaplan–Meier survival curves, and the differences between the subgroups were determined using the log-rank (Mantel–Cox) test for *p*-values and the Cox regression analysis for hazard ratios (HRs). Independent baseline characteristics from the univariate Cox regression analysis with *p*-values below 0.05 were included in a multivariate Cox regression analysis along with the expression of significant plasma proteins. Differences in the expressional dynamics over time were measured using Tukey’s multiple comparison test. The correlation between the expression of two proteins was determined using a linear regression, and differences in the baseline expressions were calculated using Welch’s *t*-test. The association between binary variables was calculated with Fisher’s exact test. *p*-values below 0.05 were considered significant.

The statistical analyses were conducted in GraphPad Prism version 9.5.1 (GraphPad Software, San Diego, CA, USA) and R version 4.2.1 (R Foundation for Statistical Computing, Vienna, Austria).

## 3. Results

### 3.1. Patients

A total of 42 patients with advanced NSCLC were included in this study. All included patients had a baseline blood sample available (T_0_). A total of 37 patients (88.09%) had a sample collected after one cycle of treatment (T_1_), and 36 patients (85.71%) had a sample after two cycles of treatment (T_2_). A total of 31 patients (73.81%) had available samples from both T_1_ and T_2_. The baseline characteristics for the patient cohort can be seen in Appendix A, and are as expected for patients with advanced NSCLC, with the majority being active or former smokers (97.62%) and adenocarcinomas being the major histological subtype (80.95%). Squamous cell histology was the only baseline characteristic associated with PFS (HR: 2.69, 95% confidence interval (CI): 1.06–5.95, *p* = 0.023) and OS (HR: 2.65, 95% CI: 1.04–5.99, *p* = 0.027) in the univariate Cox analyses (Appendix A). In the multivariate Cox analyses, squamous cell histology was only significantly associated with PFS (HR: 2.94, 95% CI: 1.10–7.07, *p* = 0.021), but not with OS (HR: 2.35, 95% CI: 0.91–5.45, *p* = 0.057) (Appendix A). At the time of data cutoff, 34 patients (80.95%) had progressed, four patients (9.52%) died before progression, and a total of 34 patients (80.95%) died due to any cause. The median PFS and OS rates were 7.23 (range: 1.51–51.32 months) and 21.63 (range: 1.51–54.41 months) months, respectively. The patients had a median PD-L1 TPS of 70% (range: 1–100%). Using the univariate Cox analysis, we found no significant association between a PD-L1 TPS rate above 50% and PFS (HR: 0.62, 95% CI: 0.28–1.56, *p* = 0.26) or OS (HR: 0.52, 95% CI: 0.24–1.32, *p* = 0.13) (Appendix A). Seven patients (16.67%) had *KRAS* mutations detected in a tissue biopsy before treatment initiation, and two patients (4.76%) had a *BRAF* mutation. No statistically significant association was found between *KRAS* mutations in tissue biopsies and survival (PFS: HR: 0.51, 95% CI: 0.17–1.21, *p* = 0.17 and OS: HR: 0.31, 95% CI: 0.07–0.87, *p* = 0.053). A total of 41 patients had their ctDNA analyzed using targeted NGS (with one patient being excluded due to a lack of plasma). *KRAS* and *TP53* mutations were the most commonly identified mutations in ctDNA [24]. *KRAS* mutations were identified in ctDNA from sixteen patients (39.02%) and *TP53* mutations in nineteen patients (46.34%). No association to survival was found for either *KRAS* ctDNA mutations (PFS: HR: 0.67, 95% CI: 0.33–1.30, *p* = 0.25 and OS: HR: 0.51, 95% CI: 0.23–1.05, *p* = 0.080) or *TP53* ctDNA mutations (PFS: HR: 0.71, 95% CI: 0.37–1.25, *p* = 0.30 and OS: HR: 1.19, 95% CI: 0.60–2.35, *p* = 0.62).

### 3.2. Expression of Plasma Proteins

Plasma samples from patients with advanced NSCLC were analyzed with Olink PEA to examine the expression levels of 92 plasma proteins (Appendix A). Samples obtained before treatment initiation (T_0_), and after one (T_1_) and two cycles of treatment (T_2_), were included to evaluate if the plasma proteins had predictive values before treatment initiation and during the initial weeks of treatment. To investigate the association between expression levels and treatment responses, the patients were divided into two groups based on whether they had above- or below-median PFS rates (median = 7.23 months). Comparing the protein expression levels between the two groups at T_0_ showed that the Fas ligand (FASLG) and inducible T-cell co-stimulator ligand (ICOSLG) were significantly higher expressed in patients with longer PFS rates (Figure 1A). At T_1_, FASLG remained significantly higher expressed in patients with longer PFS rates, while the high expression of interleukin-6 (IL-6) was associated with a shorter PFS rate (Figure 1B). At T_2_, FASLG continued to be highly expressed in patients with longer PFS rates along with C-X-C-motif chemokine 5 (CXCL5) (Figure 1C). Furthermore, C-C motif chemokine 20 (CCL20), C-X-C motif chemokine 13 (CXCL13), mucin-16 (MUC16), and matrix metalloproteinase 12 (MMP12) were significantly highly expressed in patients with a below-median PFS rate at T_2_. Despite ICOSLG, only being higher expressed in patients with an above-median PFS at T_0_, there was a trend towards patients with above-median PFS rates having higher levels of ICOSLG at T_2_, which was on the threshold of significance (*p* = 0.051), while no trend was observed at T_1_ (*p* = 0.29).

### 3.3. FASLG and ICOSLG as Predictive Biomarkers

The high expressions of both FASLG and ICOSLG in patients with longer PFS rates at T_0_ led us to investigate the predictive value of the two plasma proteins with the Kaplan–Meier survival analysis. The median expressions of ICOSLG and FASLG, respectively, were used to stratify patients into subgroups. Patients with an ICOSLG expression above the median had a significantly longer PFS (9.73 vs. 5.33 months, HR: 0.51, 95% CI: 0.26–1.00, log-rank *p* = 0.049) and OS (28.60 vs. 19.76 months, HR: 0.44, 95% CI: 0.21–0.89, log-rank *p* = 0.022) (Figure 2A,B) compared to patients with an ICOSLG expression below the median. Using the Cox univariate analysis with ICOSLG expression as a continuous variable, a significant association with PFS (HR: 0.21, 95% CI: 0.06–0.79, *p* = 0.022) and OS (HR: 0.23, 95% CI: 0.06–0.87, *p* = 0.030) was identified. Stratifying patients according to median FASLG expression showed that patients with an expression above the median had a tendency toward a longer PFS (10.45 vs. 4.83 months, HR: 0.63, 95% CI: 0.33–1.20, log-rank *p* = 0.15) and OS (27.72 vs. 19.59 months, HR: 0.63, 95% CI: 0.32–1.26, log-rank *p* = 0.19); however, this was not significant (Figure 2C,D). Stratifying based on the median expression may, however, not be the most biologically relevant cutoff point; thus, we tested whether a dose–response relationship was present between the level of FASLG in quartiles and PFS or OS (Appendix A). The difference between the quartile subsets was statistically significant (2.56 vs. 13.28 vs. 5.87 vs. 23.67 months, log-rank *p* = 0.014 for PFS and 41.95 vs. 37.78 vs. 20.25 vs. 51.32 months, log-rank *p* = 0.008 for OS), with patients with the highest quartile of FASLG expression having the longest PFS and OS rates. Furthermore, we conducted a Cox univariate analysis with FASLG as a continuous variable showing a significant association with PFS (HR: 0.42, 95% CI: 0.22–0.82, *p* = 0.009) and OS (HR: 0.38, 95% CI: 0.19–0.76, *p* = 0.005). Based on this, we chose patients with the highest FASLG expression (which we defined as within the upper quartile) compared with the rest of the patients. As shown in Figure 2E,F, this demonstrated that the patients with the highest level of FASLG at T_0_ had significantly longer PFS (23.67 vs. 4.83 months, HR = 0.34, 95% CI: 0.14–0.73, log-rank *p* = 0.006) and OS (41.26 vs. 19.59 months, HR: 0.30, 95% CI: 0.11–0.70, log-rank *p* = 0.007) rates compared to the rest of the patients. We compared clinical baseline characteristics between patients in the high or low FASLG and ICOSLG groups and found no differences between the two groups (Appendix A). To our surprise, we also found no statistically significant difference between the best radiologically assessed overall response in the two groups (Fisher’s exact test, *p* = 0.75).

The hierarchical clustering of the immune-related plasma proteins at all three time points was performed and is shown in the heatmap in Appendix A. This visualization allowed for the observation of patterns of protein expression among different patient samples collected at different time points. The heatmap color-codes the scaled protein expression levels, with red colors indicating a higher expression and blue colors a lower expression. The analysis revealed a large degree of variability between individual patient samples, with patients in the upper part of the heatmap having a higher expression across all the analyzed proteins, while the patients in the lower part of the heatmap had a global low expression of the proteins. Importantly, we noted a pattern where samples from the same patient at different time points tended to cluster together. This suggests that the interpatient variation is greater than the intra-patient variation over time. Furthermore, the analysis shows a close clustering of ICOSLG and FASLG. The proximity of the two proteins in the heatmap suggests a similar expression pattern of the two proteins.

A comparison of the expressions of FASLG and ICOSLG in the baseline blood sample using the F-test comparing variances showed a significant difference in the variances of the two proteins (*p* < 0.001) (FASLG: range of 3.94–6.10, standard deviation (SD): 0.50, and ICOSLG: range of 4.55–5.81, SD: 0.27). The difference between the expressions of FASLG and ICOSLG was assessed with a paired t-test where no significant difference was found (*p* = 0.32) (Figure 3A), demonstrating that the two proteins had similar mean expression values despite being significantly different in variance.

The linear regression analysis showed a significant correlation between the expressions of FASLG and ICOSLG (R^2^ = 0.14 and *p* = 0.015) (Figure 3B). Looking at the dynamics in the initial weeks of treatment, FASLG and ICOSLG differ with respect to their changes over time. FASLG shows a significant increase from T_0_ to both T_1_ and T_2_ (*p* < 0.001 and *p* = 0.014) (Figure 3C). In contrast, ICOSLG shows minor changes over time and little variation between patients (T_0_ to T_1_: *p* = 0.94 and T_0_ to T_2_: *p* = 0.86) (Figure 3D). These data point to the fact that, even though high FASLG and ICOSLG expressions are both associated with superior survival, and the protein expressions are positively correlated at T_0_, the two proteins display diverse interpatient variance values and dynamics during treatment.

Since the high expressions of both FASLG and ICOSLG showed an association towards longer PFS and OS rates, we wished to investigate if we could combine the expression profiles. Based on this, we characterized four patient subgroups: FASLG_low_/ICOSLG_low_, FASLG_high_/ICOSLG_low_, FASLG_low_/ICOSLG_high_, and FASLG_high_/ICOSLG_high_. No significant difference was observed between the groups; however, FASLG_low_/ICOSLG_low_ showed shorter PFS (4.49 vs. 11.21 vs. 16.93 vs. 9.11 months, log-rank *p* = 0.11) and OS (18.0 vs. 19.86 vs. 21.80 vs. 28.77 months, log-rank *p* = 0.065) rates compared to the rest of the groups that had high expressions of either FASLG, ICOSLG, or both (Appendix A). Based on this finding, we characterized a subgroup of patients with an above-median expression of FASLG and/or ICOSLG at T_0_ (*n* = 28) as the high-expression subgroup, and a subgroup with a low expression of both proteins at T_0_ (*n* = 14) as the low-expression subgroup. We found that patients in the high-expression subgroup had significantly longer PFS (10.83 vs. 4.49 months, HR: 0.43, 95% CI: 0.21–0.89, log-rank *p* = 0.017) and OS (27.13 vs. 18.0 months, HR: 0.39, 0.19–0.83, log-rank *p* = 0.011) rates than patients in the low-expression subgroup (Figure 4A,B). The HRs for the high-expression subgroup (*n* = 28) were similar to the HRs calculated for the patient subgroup with an FASLG expression in the upper quartile (*n* = 11) (HR for PFS: 0.43 vs. 0.34 (95% CI: 0.21–0.89 vs. 0.14–0.73), HR for OS: 0.39 vs. 0.30 (95% CI: 0.19–0.83 vs. 0.11–0.70)) (Figure 4C,D). This shows that this larger combined subgroup can be used in the following analyses without compromising the predictive value.

### 3.4. FASLG and ICOSLG Expressions Combined with ctDNA Mutations Are Associated with Progression-Free Survival

In a previously published study with the same patient cohort (*n* = 41), excluding one patient due to a lack of plasma, we demonstrated that the presence of ctDNA mutations after one or two cycles of treatment (T_x_) was associated with shorter PFS and OS rates [24]. We investigated whether there was an association between the expressions at T_0_ of FASLG and ICOSLG and the detection of ctDNA mutations after treatment initiation. We found no correlation between the two variables (*p* > 0.99) (Figure 5A). Due to the variables not being associated, we added the ctDNA mutation status (present or absent at T_x_) to the two-patient subgroups identified in Figure 4. The patients in the high-expression subgroup were stratified by the detection of ctDNA mutations at T_x_. This analysis showed that the two variables could supplement each other and patients with high ICOSLG and/or FASLG expressions and no detectable ctDNA mutations had a significantly longer PFS rate (25.71 vs. 6.35 months, HR: 0.33, 95% CI: 0.12–0.81, log-rank *p* = 0.015) (Figure 5B) than patients in the high-expression subgroup with detectable ctDNA mutations. We also observed a tendency towards longer OS; however, this was not significant (34.62 vs. 17.00 months, HR = 0.45, 95% CI: 0.16–1.15, log-rank *p* = 0.099) (Figure 5C). The same trend was observed for PFS when stratifying patients in the low-expression subgroup according to their ctDNA mutation status (11.72 vs. 4.09 months, HR: 0.25, 95% CI: 0.038–1.00, log-rank *p* = 0.063) and OS (22.16 vs. 15.08 months, HR = 0.69, 95% CI: 0.18–2.16, log-rank *p* = 0.53): however, the difference was not significant, which could be explained by the small subgroups in this analysis (Appendix A). Finally, we wanted to compare the group with the most favorable biomarker profile (high expressions of FASLG and ICOSLG and no ctDNA mutations after treatment initiation, *n* = 9) with the rest of the patients (*n* = 32). The analysis showed that the combined biomarker profile was highly predictive of treatment response in terms of PFS (25.71 vs. 4.52 months, HR: 0.28, 95% CI: 0.10–0.63, log-rank *p* = 0.004) and OS (34.62 vs. 18.0 months, HR: 0.33, 95% CI:0.12–0.77, log-rank *p* = 0.008) (Figure 5D,E).

## 4. Discussion

The use of immunotherapy in the treatment of advanced NSCLC has led to improvements in disease control and survival outcomes. PD-L1 expression in tumor tissue is presently being used as a biomarker for treatment eligibility: however, new biomarkers are needed to better identify patients who will benefit from treatment with anti-PD-1/PD-L1 immunotherapy. With the recent advances in liquid biopsies, we aimed to identify the predictive biomarkers in plasma samples from patients with advanced NSCLC treated with pembrolizumab.

Using Olink’s Immuno-Oncology panel, we performed a broad screening of 92 different proteins. The proteins present in the plasma may have originated from active secretion or release due to tissue damage or cell death [25]. We identified that FASLG and ICOSLG levels at baseline and the presence of ctDNA mutations after treatment initiation had a potential predictive value for treatment response.

Patients with above-median PFS rates had significantly higher FASLG and ICOSLG expressions than patients with shorter PFS rates before treatment initiation. Parallel to this finding, baseline levels of ICOSLG and FASLG could be used to identify patients with longer PFS and OS rates in the Kaplan–Meier survival analyses. This was performed using thresholds at the median and upper quartile for ICOSLG and FASLG, respectively. We observed that patients with an FASLG expression in the upper quartile had a longer survival rate compared to the other quartiles. Curiously, we observed that patients with an FASLG expression in the second quartile had the second-longest survival time in the Kaplan–Meier survival analyses for PFS and OS rates. The dose–response relationship between FASLG expression and survival should be investigated in a larger cohort to choose the optimal threshold for a high FASLG expression.

When the expressions of both FASLG and ICOSLG were used in a combined model, it was demonstrated that patients with low baseline expressions of both proteins had inferior survival rates. Furthermore, as observed in the baseline sample, patients with longer PFS rates also had significantly higher expressions of FASLG after one and two cycles of therapy compared to patients with shorter PFS rates. Surprisingly, we did not find an association between high FASLG and ICOSLG expressions and the favorable best overall response determined by radiology and the RECIST criteria. It is worth noticing that ten out of the 42 patients (23.81%) had progressive disease determined at their first radiologically assessment after pembrolizumab treatment initiation. Tumors can appear to grow at the start of immunotherapy treatment, even though the treatment is working. This growth may not be true progression, but occurs due to the infiltration of immune cells in the tumor. This phenomenon is known as pseudoprogression and can lead to a misclassification of patients and the wrongful termination of pembrolizumab treatment [26,27,28]. We theorized that, if some of the patients experiencing early radiologically assessed progressive disease in reality were cases of pseudoprogression, this could have introduced bias into our results and could explain why no association was found when looking at radiologically assessed response. Alternatives to the RECIST criteria have been proposed for the radiological assessment of response to immunotherapy, such as the immune-related response criteria (irRC) and the immune RECIST (iRECIST), but are not part of the standard clinical practice [29,30,31].

FASLG is expressed on cytotoxic T cells, where it is an inducer of apoptosis through its binding with the Fas receptor [32]. Previous studies have found an association between FASLG expression and anti-cancer activity [33,34]. A recent study by Gunnarsdottir et al. using Olink PEA showed an association between high serum FASLG levels and improved OS and PFS rates in breast cancer [34], which is similar to the results shown in this study. However, a dual role of FASLG has been proposed, where cells in the tumor milieu can express FASLG to facilitate immune escape by engaging the Fas receptor on T cells leading to the apoptosis of T cells [35,36]. This can explain the effect observed in studies on different cancer types that found an association between soluble FASLG and poor prognosis [37,38,39]. In the present study, we observed that patients with favorable survival rates had high levels of soluble FASLG both before and during the initial part of treatment. A study by O’Reilly et al. found that only membrane-bound FASLG, and not the soluble counterpart, was essential for T-cell-mediated cytotoxicity [40]. We speculated that the high expression of soluble FASLG in the plasma observed in patients with longer survival rates could be a reflection of the level of membrane-bound FASLG, and thus a surrogate marker for immune activity. The high expression of soluble FASLG may therefore be a sign of an active immune system, and thus a positive predictor of anti-cancer response.

ICOSLG is expressed on somatic cells where it functions as a co-stimulatory signal for T-cell proliferation and cytokine secretion when bound to the T-cell-specific receptor ICOS [41]. Earlier studies have shown contrasting results of the association between ICOSLG and survival. A study by Holst et al. examining the expression of plasma proteins in glioblastomas found an association between the high expression of plasma ICOSLG and longer OS rates [42]. In contrast, studies on melanoma and breast cancer showed that ICOSLG expression on tumor cells promoted the expansion of regulatory T cells, and thereby immune evasion [43,44]. These differences may be due to the tissue-specific functions of ICOS/ICOSLG signaling or soluble ICOSLG having different functions than membrane-bound ICOSLG.

In this study, we found that high levels of soluble FASLG and ICOSLG before treatment initiation were predictive of longer PFS and OS rates. The contrasting results achieved in other studies indicate that we still lack a full understanding of the mechanisms of these immune-related plasma proteins. FASLG and ICOSLG are just two proteins out of a large repertoire of soluble immune-related proteins, and in future studies, it will be interesting to combine the expression of additional proteins into a more precise model for immune activity. The unadjusted *p*-values were reported in this study, due to the exploratory nature of the study. Despite not adjusting for multiple testing, we found that the level of FASLG was significant at all three time points, and the level of ICOSLG was on the threshold of significance after two cycles of treatment (*p* = 0.051) along with being significant before treatment initiation. Based on this, we believe that the risk of FASLG and ICOSLG being significant only due to multiple testing is small, but should nonetheless be validated further.

We included ctDNA in our model since our previous study showed that it was predictive of longer survival outcomes [24]. No association was observed between the absence of ctDNA mutations after treatment initiation and having high expressions of FASLG and ICOSLG at baseline. In the present study, we combined ctDNA with plasma protein expression and, therefore, were able to observe an even better distinction between patients with a good clinical outcome and those who progressed early. This combined model was advantageous since these biomarkers reflected different areas of interest concerning anti-cancer treatment. The plasma proteins were indicative of immune response, while the ctDNA dynamics both reflected the molecular composition of the tumor, the tumor burden, as well as the effect of treatment on tumor cells.

The limitations of this study included the number of patients, which resulted in small subgroups in our analyses. Despite this, the stratification led to significant differences in survival outcomes. Furthermore, the limited number of patients prevented the formation of a validation cohort, thus the results should be interpreted with caution and should be confirmed in a larger cohort to increase the statistical power and generalizability of the findings. Nevertheless, this study described patients with NSCLC in a real-world setting and provided evidence of a minimally invasive way to identify predictive biomarkers for response to immune checkpoint inhibitors.

## 5. Conclusions

This study showed that the expressions of FASLG and ICOSLG were predictive of longer survival rates in patients with advanced NSCLC treated with pembrolizumab. Furthermore, the expression of these proteins can be combined with the absence of ctDNA mutations after treatment initiation to identify a patient subgroup with favorable survival outcomes.

## Figures and Tables

**Figure 1 cancers-15-05628-f001:**
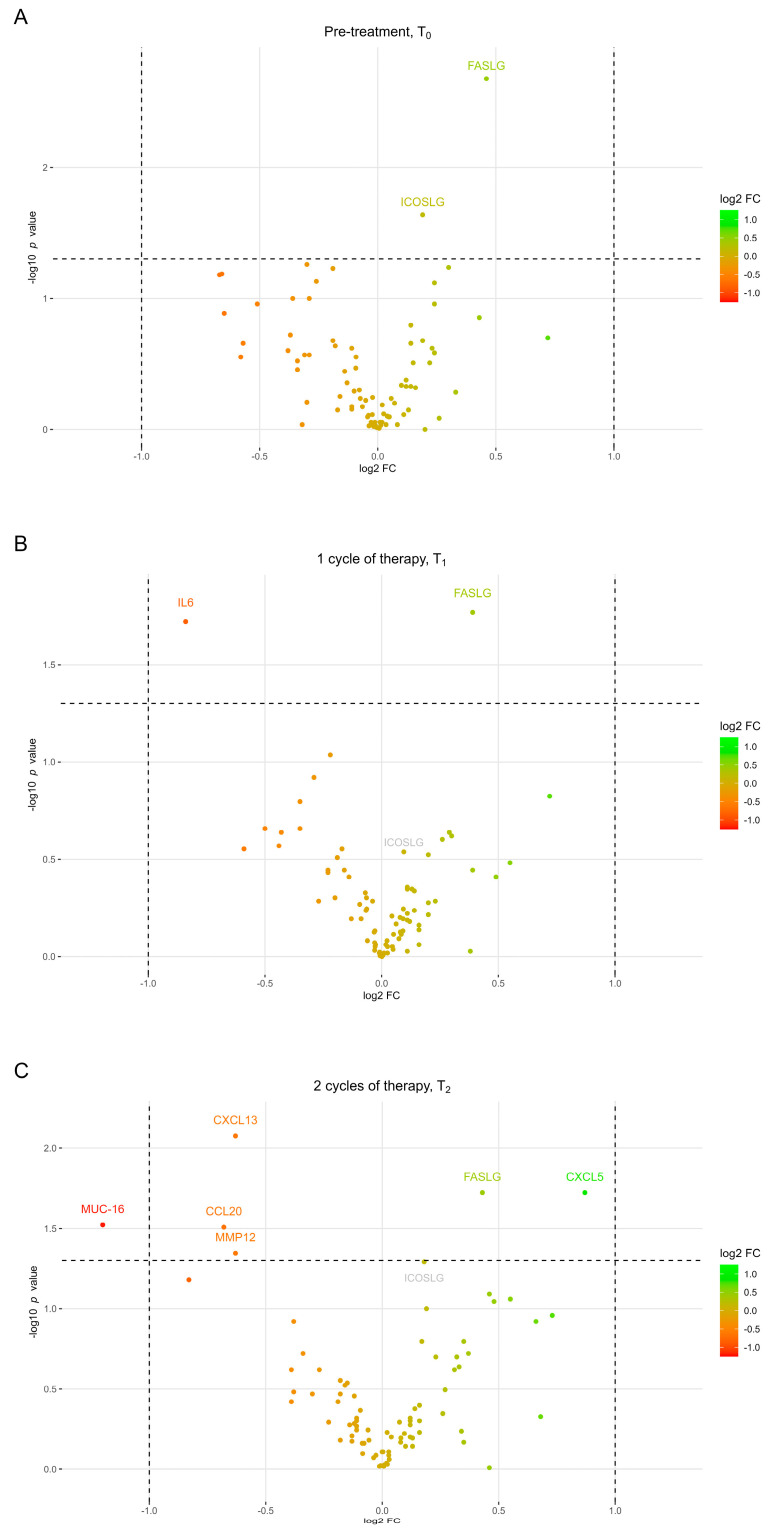
Volcano plots showing log2 fold change in plasma protein expression between patients with above-median PFS and patients with below-median PFS at three different time points. Proteins with a positive log2 FC (the right part of the volcano plot) are upregulated in patients with above-median PFS, while proteins with a negative log2 FC (the left part of the volcano plot) are upregulated in patients with below-median PFS. (**A**) Differences in protein expression before treatment initiation, (**B**) after one and (**C**) two cycles of treatment. Vertical dashed lines represent 2-fold change, while the horizontal dashed line represent a *p*-value of 0.05. Proteins with expressions below the limit of detection were excluded from the analysis. FC, fold change; PFS, progression-free survival; T_0_, pre-treatment; T_1_, after one cycle of therapy; T_2_, after two cycles of therapy.

**Figure 2 cancers-15-05628-f002:**
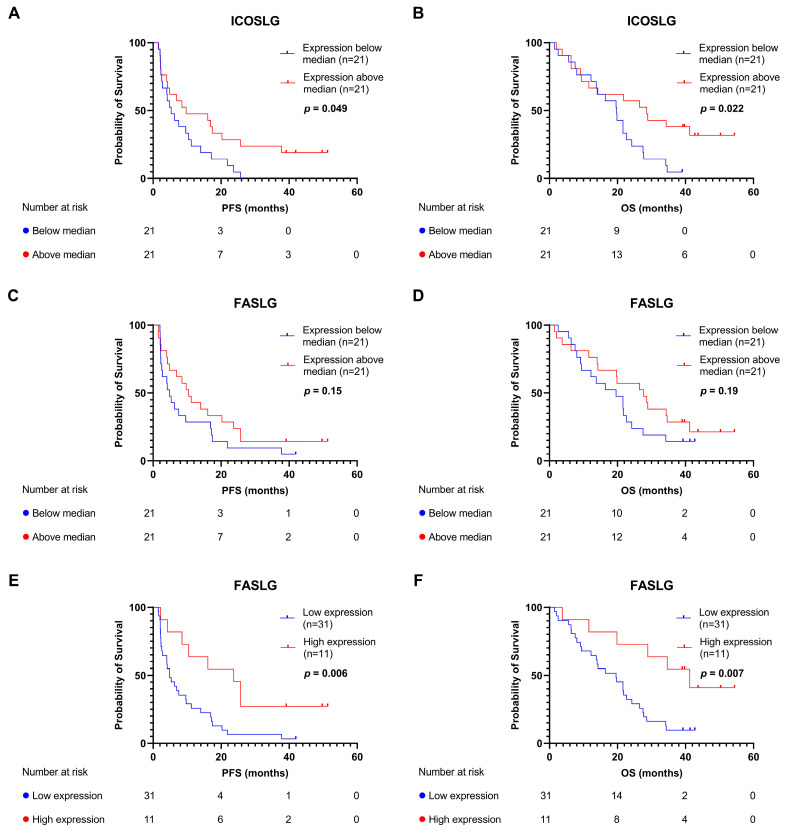
Kaplan–Meier survival analyses for PFS and OS. (**A**) PFS according to median expression of ICOSLG. (**B**) OS according to median expression of ICOSLG. (**C**) PFS according to median expression of FASLG. (**D**) OS according to median expression of FASLG. (**E**) PFS according to high expression of FASLG (expression in the upper quartile). (**F**) OS according to high expression of FASLG. FASLG, Fas ligand; ICOSLG, inducible T-cell co-stimulator ligand; OS, overall survival; PFS, progression-free survival.

**Figure 3 cancers-15-05628-f003:**
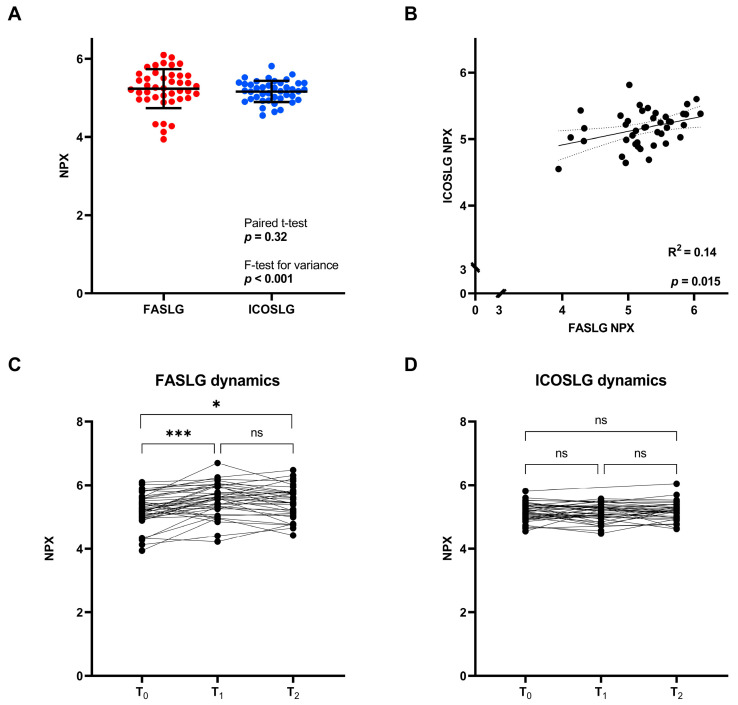
Expressions of FASLG and ICOSLG. (**A**) NPX values at T_0_. (**B**) Correlation between FASLG and ICOSLG NPX values at T_0_. The dotted lines represent the 95% confidence interval. (**C**) Dynamics in FASLG expression during the initial weeks of treatment. (**D**) Dynamics in ICOSLG expression during the initial weeks of treatment. FASLG, Fas ligand; ICOSLG, inducible T-cell co-stimulator ligand; NPX, normalized protein expression; T_0_, pre-treatment; T_1_, after one cycle of therapy; T_2_, after two cycles of therapy; ns, not significant; *, *p* < 0.05; ***, *p* < 0.001.

**Figure 4 cancers-15-05628-f004:**
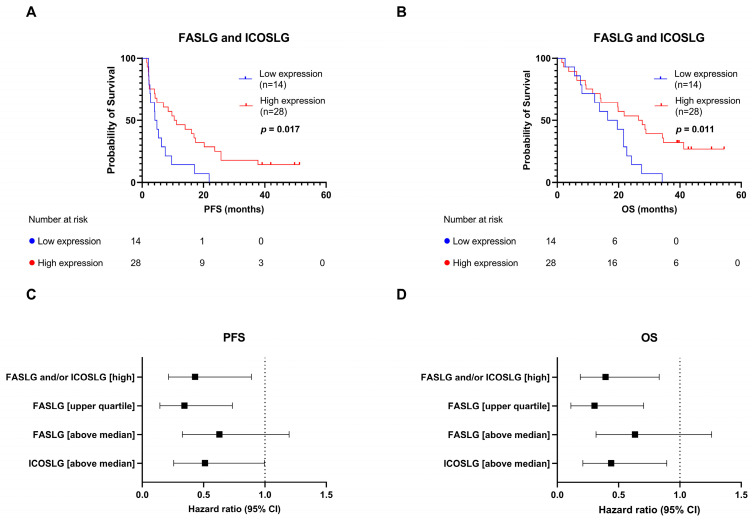
Combining expressions of FASLG and ICOSLG before treatment initiation. (**A**) PFS according to FASLG and ICOSLG expressions. Patients were stratified into the high-expression subgroup if they had an above-median expression of either FASLG and/or ICOSLG. Patients with low expressions of both proteins were stratified into the low-expression subgroup. (**B**) OS according to FASLG and ICOSLG expressions. (**C**) Forest plot showing HR and 95% CI for single and combined protein-expression subgroups for PFS. (**D**) Forest plot showing HR and 95% CI for single and combined protein-expression subgroups for OS. CI, confidence interval; FASLG, Fas ligand; ICOSLG, inducible T-cell co-stimulator ligand; OS, overall survival; PFS, progression-free survival.

**Figure 5 cancers-15-05628-f005:**
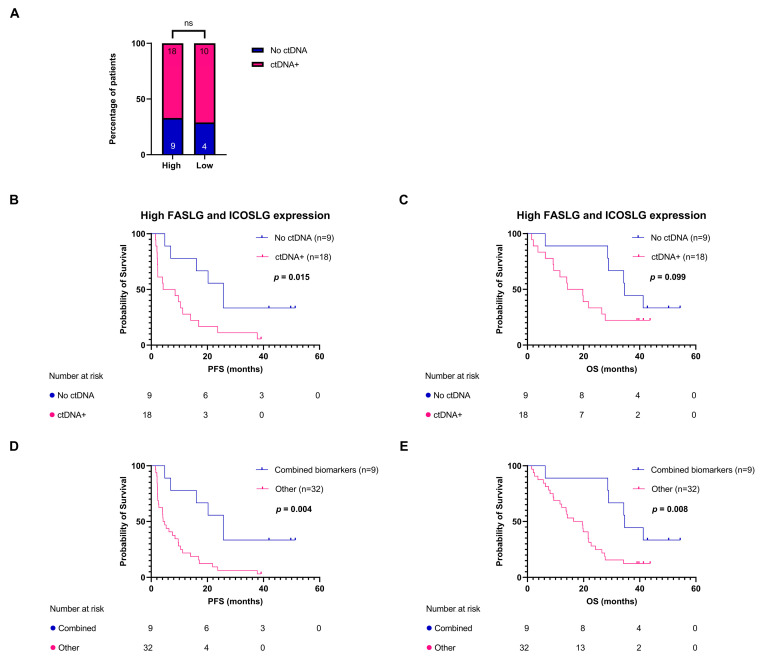
Combining circulating biomarkers. (**A**) Proportion of patients in each FASLG and ICOSLG expression subgroup having a presence or absence of ctDNA mutations after treatment initiation. (**B**) PFS according to presence or absence of ctDNA mutations after treatment initiation in the high-expression FASLG and ICOSLG subgroups. (**C**) OS according to the presence or absence of ctDNA mutations after treatment initiation in the high FASLG and ICOSLG expression subgroup. (**D**) PFS according to a combined biomarker profile, i.e., patients in the high-expression FASLG and ICOSLG subgroups with an absence of ctDNA mutations, compared to the rest of the patients. (**E**) OS according to a combined biomarker profile. FASLG, Fas ligand; ICOSLG, inducible T-cell co-stimulator ligand; ns, not significant; OS, overall survival; PFS, progression-free survival.

## Data Availability

Research data are available upon reasonable request to the corresponding author.

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
