# Peer review of "Plasma Immune Proteins and Circulating Tumor DNA Predict the Clinical Outcome for Non-Small-Cell Lung Cancer Treated with an Immune Checkpoint Inhibitor"

_cancers, 2023, doi:10.3390/cancers15235628_

Round 1

Reviewer 1 Report

Comments and Suggestions for Authors

In the present paper, Stensgaard et al. perform a multiplexed protein profiling of plasma samples in a limited (n=42) cohort of NSCLC patients treated with pembrolizumab. Applying a commercial platform, they identify

 FASLG and ICOSLG as potential prognostic biomarkers in this setting.

Since the same samples have been previously included in another study with a similar design, namely immunotherapy biomarkers discovery, and the approach is broad, rather than targeted, I think that some aspects should be further elucidated.

Patient characteristics. What was the Molecular status of the tumors? Any association with PFS/OS?

I miss a table describing the association between the two key biomarkers FASLG and ICOSLG and patient and tumor characteristics.

The cut off for FASLG has been chosen at Q4, and a biological rational is proposed. However, from Suppl Fig S2 it looks like Q2 and Q4 FALSG quartiles have a similar better prognosis than Q1 and Q3. How do the au explain this finding?

Fig 3 is difficult to interpret. I would suggest to move it as Suppl. 

Comments on the Quality of English Language

The quality is good

Author Response

Reviewer 1

We thank the reviewer for taking the time to review this manuscript. Please find the detailed responses below and the corresponding revisions highlighted in red in the re-submitted files.

Comment 1: Patient characteristics. What was the Molecular status of the tumors? Any association with PFS/OS?

Reply 1: Thank you for the suggestion. Detection of mutations in the tissue biopsy was done per local practice as part of the standard procedure for lung cancer patients at Aarhus University Hospital. KRAS mutations were detected in seven patients and BRAF mutations in two patients. We found no association between KRAS mutations in tissue biopsies and PFS or OS. For the result of the genomic testing of ctDNA, before and after treatment initiation, we have included information on KRAS and TP53 mutations, as these were the only common mutations found in a substantial number of patients to allow for general conclusions (more than five patients). KRAS mutations were identified in sixteen patients and TP53 mutations in nineteen patients. No association was found between KRAS and TP53 mutations in ctDNA and PFS or OS. In addition, we have referenced our previous paper describing the molecular characterization of the patient cohort in more detail (Stensgaard et al.: Blood tumor mutational burden and dynamic changes in circulating tumor DNA predict response to pembrolizumab treatment in advanced non-small cell lung cancer, doi: 10.21037/tlcr-22-818.)

Changes in the text: Detection of mutations in tissue biopsies is added to Materials and Methods page 2 line 80-82. The results for the association of KRAS mutations in tissue/ctDNA and PFS and OS is added at page 4, line 195 to page 5, line 206 and Supplementary Table S2.

Comment 2: I miss a table describing the association between the two key biomarkers FASLG and ICOSLG and patient and tumor characteristics.

Reply 2: We have added columns describing baseline characteristics for the high and low FASLG and ICOSLG cohorts in Supplementary Table S1, as well as conducting Fisher’s exact test, to investigate if there were any differences in baseline characteristics between the cohorts. No statistically significant differences were identified.

Changes in the text: Added columns in Supplementary Table S1 for the FASLG and ICOSLG high or low cohorts, as well as a column containing p values, as well as a section regarding the clinical baseline characteristics at page 7, line 270-274.

Comment 3: The cut off for FASLG has been chosen at Q4, and a biological rational is proposed. However, from Suppl Fig S2 it looks like Q2 and Q4 FALSG quartiles have a similar better prognosis than Q1 and Q3. How do the au explain this finding?

Reply 3: We agree that this is indeed a curious finding. We have conducted a Cox univariate analysis for the association between FASLG expression as a continuous variable in addition to the already reported categorical variable (above median or upper quartile). We find that FASLG expression as a continuous variable is associated with PFS (p = 0.009) and OS (p = 0.005). Based on this we hypothesize that the more extreme FASLG values in Q1 and Q4 are affecting the results, while the more similar values in Q2 and Q3 does not affect the groups as much. Looking at the variance in FASLG NPX values in each group, we also observe a larger variance for the Q1 and Q4 groups (Q1: 0.148, Q2: 0.004, Q3: 0.009, Q4: 0.029). Comparing the patient group with Q4 FASLG NPX values with the rest gave significant p values for both PFS (p = 0.006) and OS (p = 0.007), while comparing Q2 with the combined Q1 and Q3 groups did not yield significant differences for neither PFS (p = 0.13) nor OS (p = 0.08). To summarize, we hypothesize that it is the more extreme expression values that affect the survival of the patients. The Q2 and Q3 cohort are more uniform in FASLG NPX values compared to Q1 and Q4. The difference in survival between the Q2 and Q3 cohort was not statistically significant (p = 0.13). However, we acknowledge that this should be investigated further in a larger cohort and have added this to the Discussion.

Changes in the text: The results of the Cox univariate analysis for association of FASLG as a continuous variable is shown at page 7, line 262-265. A section have been added to the Discussion regarding larger cohorts being needed to choose the optimal threshold for high FASLG expression (Page 12, line 405-411).

Comment 4: Fig 3 is difficult to interpret. I would suggest to move it as Suppl.

Reply 4: We thank the reviewer for the suggestion. We have added a few additional lines further elucidating our results in the heatmap and have moved Figure 3 to Supplementary Figure S3 as suggested.

Changes in the text: Page 8, line 284 to page 9, line 296. Figure 3 has been moved to Supplementary Figure S3. The following figures and supplementary figures have, as a consequence of this, changed names.

Reviewer 2 Report

Comments and Suggestions for Authors

Stensgaard et al. present a study on the prognostic and predictive value of two plasma proteins, FASLG and ICOSLG, in patients with non-small cell lung cancer (NSCLC) treated with an immune checkpoint inhibitor. They are using the Olink Proximity Extension Assay (PEA) to measure the expression levels of 92 proteins and correlate them with clinical outcomes such as response to therapy, progression-free survival and overall survival. They also analyze the levels of circulating tumor DNA (ctDNA) as a biomarker of tumor burden and its association with survival.

The paper is well structured, clear and concise. The methodology is appropriate and robust, and the results are consistent and reliable. The authors draw reasonable conclusions from their data and acknowledge the limitations of their study. One of the main limitations is the small sample size, which reduces the statistical power and generalizability of the findings. A larger cohort of patients would be needed to confirm the validity and clinical utility of FASLG and ICOSLG as biomarkers of response to immunotherapy in non-small cell lung cancer.

Author Response

Reviewer 2

We thank the reviewer for the positive review of our manuscript. Please find the detailed response below and the corresponding revisions highlighted in red in the re-submitted files.

Comment 1: The paper is well structured, clear and concise. The methodology is appropriate and robust, and the results are consistent and reliable. The authors draw reasonable conclusions from their data and acknowledge the limitations of their study. One of the main limitations is the small sample size, which reduces the statistical power and generalizability of the findings. A larger cohort of patients would be needed to confirm the validity and clinical utility of FASLG and ICOSLG as biomarkers of response to immunotherapy in non-small cell lung cancer.

Reply 1: We thank the reviewer for the kind words and agree that the sample size is the main limitation of the paper. We hope that the significant differences in survival between the high and low FASLG and ICOSLG subgroups will lead to larger studies examining the value of FASLG and ICOSLG as biomarkers.

Changes in the text: We have further elaborated on the limitations of a small sample size at page 13, line 480-487.

Reviewer 3 Report

Comments and Suggestions for Authors

A review of Manuscript entitled " Plasma immune proteins and circulating tumor DNA predict

response to immune checkpoint inhibitor treatment in non-small cell lung cancer."

The manuscript submitted by Stensgaard et al. (2023) aimed to identify predictive biomarkers that can differentiate patients who respond to pembrolizumab from non-responders. To achieve this, the authors investigated immune-related plasma proteins and ctDNA levels. The study used the Olink Target 96 Immuno-Oncology panel (Olink Proteomics) to analyze 92 plasma proteins among patients grouped based on median time to progression (PFS) and median overall survival (OS). The authors evaluated three potentially predictive proteins: Fas ligand (FASLG), inducible T cell costimulator ligand 26 (ICOSLG), and interleukin-6 (IL-6). Higher levels of FASLG and ICOSLG in plasma were correlated with longer PFS and OS in different subgroups, while higher levels of IL-6 were associated with shorter PFS. In addition, the authors combined the absence of detectable ctDNA in plasma with these protein levels and identified a patient subgroup with favorable survival. The authors used several statistical tests to demonstrate and prove possible associations. Overall, the authors presented new and interesting data; however, I have some major comments, and some minor improvements are still necessary.

Major comments:

1)     The aim of the paper can be misleading. The authors indicate that the main aim of the study was to investigate the immune-related plasma proteins and ctDNA level in order to identify predictive biomarkers that can distinguish patients that respond to pembrolizumab from non-responders (Introduction, line 71). However, in this study all the analysis are based on PFS and OS. There is luck of information about the method of patients categorization as responders or non-responders (how many patients had a partial response (PR), complete response (CR), progressive disease (PD) or stable disease (SD) and appropriate statistical analysis. I would like to ask the authors for a comment on this issue.

2)     Results, 3.2 paragraph “Expression of plasma proteins”, line 170:To investigate the association between expression levels and treatment response, patients were divided into two groups based on whether they had above or below median PFS (median=7.23 months).” Please clarify why the medians of PFS and OS were choose as cut-off points for treatment response? Additionally, I suggest using univariate Cox proportional hazards model analysis to determine the relationship between protein level (a continuous variable) and clinical endpoints PFS and OS.

3)     Material and Methods, 2.3 paragraph “Olink Proximity Extension Assay”, line 95: there is luck of even shortcut of protein level analysis method (Olink Target 96 Immuno-Oncology panel), which is the core of the paper. The reference [25] (line 100): Olink. Olink Target 96 Immuno-Oncology. Available online: https://www.olink.com/products-services/target/immuno-oncology/ (line 512)“ can be change or delated. Additionally, please indicate briefly what is the NPX and how it is calculated. Using the phrases “protein expression” in the text without clarification may be misleading for some readers.

Minor comments:

1.     Abstract, line 24 “In plasma samples we quantified the level of 92 proteins as well as 24 circulating tumor DNA (ctDNA).” Could you please specify the protein analysis method that was utilized?

2.     Material and Methods, 2.2 paragraph “Sample Collection and Preparation”, line 91:Plasma samples taken before treatment initiation (T0) and after one (T1) and two cycles of treatment (T2) were chosen for analysis”. Could you please specify the time intervals between To, T1 and T2?

3.     Material and Methods, 2.3 paragraph “Olink Proximity Extension Assay”, line 100: The samples were analyzed using the Olink Target 96 Immuno-Oncol-98 ogy panel (Olink Proteomics), which enables the detection of 92 plasma proteins related 99 to immuno-oncology [25]. Please add the reference to Table S3 (“Proteins in the Olink Target 96 Immuno-Oncology Panel)”.

4.     Material and Methods, 2.4 paragraph “Next-Generation Sequencing”, line 103: The Next-Generation Sequencing method used in this paper is not related to the results presented. To make the paper more informative, I suggest adding a description of the isolation and measurement method for ctDNA at this point. Furthermore, it would be highly beneficial to include a table showing the correlation between ctDNA levels and protein levels.

5.     Material and Methods, 2.5 paragraph “Statistical Analysis”, line 120: “The fold change in expression was calculated on a linear scale as the geometric mean of the first group divided by the geometric mean of the 121 second group.” What the first and second groups means?

6.     Figure 1, line 186:  “Volcano plots showing log2 fold change in plasma protein expression between patients with above median PFS and patients with below median PFS…” Figure 1 would be more readable if authors showed the PFS scale with the indication of median value on graphs.

7.     Results, 3.2 paragraph “Expression of plasma proteins”, line 204:” Stratifying patients according to median FASLG expression showed that patients with an expression above the median had a tendency toward longer PFS (10.45 vs. 4.83 months, HR: 0.63, 95% CI: 0.33-1.20, log-rank p = 0.15) and OS (27.72 vs. 19.59 months, HR: 0.63, 206 95% CI: 0.32-1.26, log-rank p = 0.19), however, not significant (Figure 2C and 2D).” These results are not significant. To clarify the manuscript, the authors could shorten these descriptions.

8.     Figure 3, line237: “Heatmap showing scaled protein expression for the patients at all three time”. Due to the large amount of data and time points analyzed together, it is difficult to discern the conclusion presented in the text: line 233: “Furthermore, the analysis shows that ICOSLG and FASLG cluster together in the heatmap suggesting a similar expression pat-234 tern of the two proteins”. Can you elucidate it?

9.     Results, 3.3 paragraph “FASLG and ICOSLG as Predictive Biomarkers”, line 271:” However, the FASLGlow/ICOSLGlow showed shorter PFS (4.49 vs. 271 11.21 vs. 16.93 vs. 9.11 months) and OS (18.0 vs. 19.86 vs. 21.80 vs. 28.77 months) compared to the rest of the groups that had high expression of either FASLG, ICOSLG or both.” Are these results statistically relevant?

Author Response

Reviewer 3

We thank Reviewer 3 for the thorough review of the manuscript. Please find the detailed response below and the corresponding revisions highlighted in red in the re-submitted files.

Major comments:

Comment 1: The aim of the paper can be misleading. The authors indicate that the main aim of the study was to investigate the immune-related plasma proteins and ctDNA level in order to identify predictive biomarkers that can distinguish patients that respond to pembrolizumab from non-responders (Introduction, line 71). However, in this study all the analysis are based on PFS and OS. There is luck of information about the method of patients categorization as responders or non-responders (how many patients had a partial response (PR), complete response (CR), progressive disease (PD) or stable disease (SD) and appropriate statistical analysis. I would like to ask the authors for a comment on this issue.

Reply 1: This is a valid point, and we have conducted a Fisher’s exact test to show the association between the high expression FASLG and ICOSLG group and radiologically determined response. Furthermore, we have added information about best overall response to Supplementary Table S1, showing baseline characteristics. We find no association between high FASLG and ICOSLG expression and radiological response, which we have commented on in our revised Discussion section.
We acknowledge that the term ‘responders’ can be misleading, and we have rephrased our aim from identifying responders to predicting clinical outcome.

Changes in the text: We added the definition of best overall response to page 3, line 141-143. Rephrasing of aim at page 2, line 73-74, and the use of the term responders at page 1, line 34-35 and page 13, line 474-475, as well as adding best overall response values to Supplementary Table S1. We added a section to the Discussion regarding radiologically assessed response to immunotherapy at page 12, line 416-430 along with the references 29-34.

Comment 2: Results, 3.2 paragraph “Expression of plasma proteins”, line 170: “To investigate the association between expression levels and treatment response, patients were divided into two groups based on whether they had above or below median PFS (median=7.23 months).” Please clarify why the medians of PFS and OS were choose as cut-off points for treatment response? Additionally, I suggest using univariate Cox proportional hazards model analysis to determine the relationship between protein level (a continuous variable) and clinical endpoints PFS and OS.

Reply 2: We chose to use PFS as a clinical endpoint in our analysis since this marker includes a time perspective in contrast to objective response rate, which reports the best assessed radiological response. Using PFS as a clinical endpoint gave us the possibility to look at a continuous variable which contains information about patients who achieve long-lasting effect of pembrolizumab treatment. We wanted to investigate if patients with longer PFS had a different profile of immune proteins in the blood. Thus, we divided based on the median PFS to be able to analyze two patient subgroups of reasonable size. In the Kaplan-Meier survival analysis for OS, we saw that the proteins identified using median PFS, also was associated with OS.
We agree with the reviewer, that an analysis using the continuous FASLG and ICOSLG expression could be a good addition to the results. We found a significant association between both proteins as continuous variables and have added these results to the text.

Changes in the text: Cox univariate proportional hazard analysis was added using continuous FASLG and ICOSLG values at page 7, line 250-252 and line 262-265.

Comment 3: Material and Methods, 2.3 paragraph “Olink Proximity Extension Assay”, line 95: there is luck of even shortcut of protein level analysis method (Olink Target 96 Immuno-Oncology panel), which is the core of the paper. The reference [25] (line 100): Olink. Olink Target 96 Immuno-Oncology. Available online: https://www.olink.com/products-services/target/immuno-oncology/ (line 512)“ can be change or delated. Additionally, please indicate briefly what is the NPX and how it is calculated. Using the phrases “protein expression” in the text without clarification may be misleading for some readers.

Reply 3: We thank the reviewer for the suggestion and have further explained the Olink proximity extension assay and elaborated on the definition of the normalized protein expression (NPX) value.

Changes in the text: Explanation of the Olink PEA method and calculation of NPX values at page 3, line 105-122.

Minor comments:

Comment 4: Abstract, line 24 “In plasma samples we quantified the level of 92 proteins as well as 24 circulating tumor DNA (ctDNA).” Could you please specify the protein analysis method that was utilized?

Reply 4: We have added information regarding the protein assay (Olink proximity extension assay) and ctDNA detection method (targeted sequencing) to the abstract.

Changes in the text: Mentioning of utilized methods in abstract at page 1, line 24-26.

Comment 5: Material and Methods, 2.2 paragraph “Sample Collection and Preparation”, line 91: “Plasma samples taken before treatment initiation (T0) and after one (T1) and two cycles of treatment (T2) were chosen for analysis”. Could you please specify the time intervals between To, T1 and T2?

Reply 5: The T0 samples were collected 0-27 days before start of pembrolizumab treatment (median 2 days before). The T1 samples were collected after 17-43 days of treatment (median 21 days after), and the T2 samples were collected after 34-90 days of treatment (median 43.5 days after).

Changes in the text: The time intervals for the sample collection have been added at page 3, line 96-99.

Comment 6: Material and Methods, 2.3 paragraph “Olink Proximity Extension Assay”, line 100: The samples were analyzed using the Olink Target 96 Immuno-Oncol-98 ogy panel (Olink Proteomics), which enables the detection of 92 plasma proteins related 99 to immuno-oncology [25]. Please add the reference to Table S3 (“Proteins in the Olink Target 96 Immuno-Oncology Panel)”.

Reply 6: The reference to the Supplementary Table S3 has been added.

Changes in the text: Reference to Supplementary Table S3 added at page 3, line 115.

Comment 7: Material and Methods, 2.4 paragraph “Next-Generation Sequencing”, line 103: The Next-Generation Sequencing method used in this paper is not related to the results presented. To make the paper more informative, I suggest adding a description of the isolation and measurement method for ctDNA at this point. Furthermore, it would be highly beneficial to include a table showing the correlation between ctDNA levels and protein levels.

Reply 7: We have added a description of the isolation and library preparation to the Next-Generation Sequencing data. Furthermore, we have clarified the definition of ctDNA negative and positive samples.

Changes in the text: We have added a longer description of the sequencing method at page 3, line 126-132. The ctDNA analysis software and definition of ctDNA positive and negative samples have been added at page 4, line 156-160.

Comment 8: Material and Methods, 2.5 paragraph “Statistical Analysis”, line 120: “The fold change in expression was calculated on a linear scale as the geometric mean of the first group divided by the geometric mean of the 121 second group.” What the first and second groups means?

Reply 8: The first and second group refer to the patient cohort with either above or below median PFS. We have clarified this in the text.

Changes in the text: Clarification of the compared groups at page 4, line 147-150.

Comment 9: Figure 1, line 186:  “Volcano plots showing log2 fold change in plasma protein expression between patients with above median PFS and patients with below median PFS…” Figure 1 would be more readable if authors showed the PFS scale with the indication of median value on graphs.

Reply 9: We thank the reviewer for the opportunity of making these results clearer. The volcano plots show the -log10 p value on the y-axis and the log2 fold change on the x-axis. The fold change is calculated for each protein as the difference between the patient group with above median PFS compared to the patient group with below median PFS. Thus, PFS is used as a binary variable to define the two groups and calculate the fold change. A positive log2 fold change indicates that the protein is upregulated in patients with above median PFS, while a negative log2 fold change indicates that the protein is upregulated in patients with below median PFS.

Changes in the text: We have added an explanation of the indication of a positive or negative log2 fold change to the legend to Figure 1 (page 6, line 230-232).

Comment 10: Results, 3.2 paragraph “Expression of plasma proteins”, line 204:” Stratifying patients according to median FASLG expression showed that patients with an expression above the median had a tendency toward longer PFS (10.45 vs. 4.83 months, HR: 0.63, 95% CI: 0.33-1.20, log-rank p = 0.15) and OS (27.72 vs. 19.59 months, HR: 0.63, 206 95% CI: 0.32-1.26, log-rank p = 0.19), however, not significant (Figure 2C and 2D).” These results are not significant. To clarify the manuscript, the authors could shorten these descriptions.

Reply 10: We agree with the reviewer that these results take up a lot of space, even though they are not significant. However, we feel that these results are important in order to understand the shift from looking at median expression to focusing on the upper quartile of FASLG expression. Furthermore, for transparency, we suggest keeping the reporting of analyses and statistical tests uniform throughout the paper. We are therefore of the opinion that the results in the manuscript should not be simplified, although we understand the wish to clarify the Results section.

Changes in the text: None

Comment 11: Figure 3, line237: “Heatmap showing scaled protein expression for the patients at all three time”. Due to the large amount of data and time points analyzed together, it is difficult to discern the conclusion presented in the text: line 233: “Furthermore, the analysis shows that ICOSLG and FASLG cluster together in the heatmap suggesting a similar expression pat-234 tern of the two proteins”. Can you elucidate it?

Reply 11: We agree with the reviewer that Figure 3 contains a large amount of data that can potentially be overwhelming. We have further elaborated on the findings of the heatmap in the text and have chosen to move the figure to the supplementary files for clarity.

Changes in the text: Elaboration of the findings in the heatmap at page 8, line 284 to page 9, line 296. Figure 3 has been moved to Supplementary Figure S3. The following figures and supplementary figures have, as a consequence of this, changed names.

Comment 12: Results, 3.3 paragraph “FASLG and ICOSLG as Predictive Biomarkers”, line 271:” However, the FASLGlow/ICOSLGlow showed shorter PFS (4.49 vs. 271 11.21 vs. 16.93 vs. 9.11 months) and OS (18.0 vs. 19.86 vs. 21.80 vs. 28.77 months) compared to the rest of the groups that had high expression of either FASLG, ICOSLG or both.” Are these results statistically relevant?

Reply 12: These results were not statistically significant (p = 0.11 for PFS and p = 0.065 for OS). We have now added these p values next to the survival data for better clarity of the significance of the results.

Changes in text: Rephrasing of sentences and addition of p values to the survival data at page 9, line 324 to page 10, line 328.

Round 2

Reviewer 1 Report

Comments and Suggestions for Authors

It seems like the Au ave adequately replied to the comments from the three reviewers

Author Response

We thank the reviewer for taking the time to review the manuscript and for their useful comments, which have improved the clarity of the manuscript.

Reviewer 3 Report

Comments and Suggestions for Authors

A review of Manuscript entitled " Plasma immune proteins and circulating tumor DNA predict

response to immune checkpoint inhibitor treatment in non-small cell lung cancer."

I would like to express my gratitude to the authors for addressing my comments and implementing the necessary corrections. However, I have steal some major comments, and some minor improvements are still necessary.

Major comments:

1)      “The aim of the paper can be misleading. The authors indicate that the main aim of the study was to investigate the immune-related plasma proteins and ctDNA level in order to identify predictive biomarkers that can distinguish patients that respond to pembrolizumab from non-responders (Introduction, line 71). However, in this study all the analysis are based on PFS and OS. There is luck of information about the method of patients categorization as responders or non-responders (how many patients had a partial response (PR), complete response (CR), progressive disease (PD) or stable disease (SD) and appropriate statistical analysis. I would like to ask the authors for a comment on this issue.”

Authors Reply 1: This is a valid point, and we have conducted a Fisher’s exact test to show the association between the high expression FASLG and ICOSLG group and radiologically determined response. Furthermore, we have added information about best overall response to Supplementary Table S1, showing baseline characteristics. We find no association between high FASLG and ICOSLG expression and radiological response, which we have commented on in our revised Discussion section.
We acknowledge that the term ‘responders’ can be misleading, and we have rephrased our aim from identifying responders to predicting clinical outcome.

Changes in the text: We added the definition of best overall response to page 3, line 141-143. Rephrasing of aim at page 2, line 73-74, and the use of the term responders at page 1, line 34-35 and page 13, line 474-475, as well as adding best overall response values to Supplementary Table S1. We added a section to the Discussion regarding radiologically assessed response to immunotherapy at page 12, line 416-430 along with the references 29-34.

Thank you for making the corrections. However, I would suggest also updating the manuscript title.

2)      “Material and Methods, 2.3 paragraph “Olink Proximity Extension Assay”, line 95: there is luck of even shortcut of protein level analysis method (Olink Target 96 Immuno-Oncology panel), which is the core of the paper. The reference [25] (line 100): Olink. Olink Target 96 Immuno-Oncology. Available online: https://www.olink.com/products-services/target/immuno-oncology/ (line 512)“ can be change or delated. Additionally, please indicate briefly what is the NPX and how it is calculated. Using the phrases “protein expression” in the text without clarification may be misleading for some readers.”

Authors Reply 3: We thank the reviewer for the suggestion and have further explained the Olink proximity extension assay and elaborated on the definition of the normalized protein expression (NPX) value. Changes in the text: Explanation of the Olink PEA method and calculation of NPX values at page 3, line 105-122.

I would like to express my gratitude to the authors for providing an extended description of the method. However, the “Material and Methods” chapter is a crucial part of a scientific work. Its purpose is to provide a detailed description of the experiments conducted so that other scientists can replicate and validate the results. I understand that the company "BioXpedia, Aarhus, Denmark" was responsible for protein analysis. However, authors must provide some basic information, for example about the amounts of input material that were analyzed. Without such information, it will be impossible for anyone to verify the results. It is worth noting that references [25] and [26] only provide a link to the manufacturer's website, which contains a brief description of the used Olink Target 96 Immuno-Oncology panel (Olink Prote-110 omics). However, it doesn't include a detailed description of the procedure used by the authors for their samples, including the amounts of reagents used, reaction conditions, etc. Typically, if the reactions are carried out following the protocol recommendation provided by the manufacturer, such information is included in the text, along with a brief description of the procedure. The goal is to give the reader a clear idea of the subsequent steps that were taken. In my opinion, instead of providing a link to an external website that may disappear in the future, the authors should include additional information related to the methodology in a supplement.

Minor comments:

1)      Material and Methods, 2.1 paragraph “Patients”, line 77: corrected sentence: “Pathological detection of mutations in tissue biopsies was done as part of the standard procedure for lung cancer patients 78 at Aarhus University Hospital.”

Was this sentence discussing the detection of pathogenic mutations? I'm not sure of the author's intent. Please clarify it.

2)      Material and Methods, 2.4 paragraph “Next-Generation Sequencing”, line 103: The Next-Generation Sequencing method used in this paper is not related to the results presented. To make the paper more informative, I suggest adding a description of the isolation and measurement method for ctDNA at this point. Furthermore, it would be highly beneficial to include a table showing the correlation between ctDNA levels and protein levels.

Authors Reply 7: We have added a description of the isolation and library preparation to the Next-Generation Sequencing data. Furthermore, we have clarified the definition of ctDNA negative and positive samples. Changes in the text: We have added a longer description of the sequencing method at page 3, line 126-132. The ctDNA analysis software and definition of ctDNA positive and negative samples have been added at page 4, line 156-160

Line 122 in revised manuscript: Sample preparation and sequence analysis of ctDNA was performed as part of a previously published study [24]. Briefly, cell-free DNA was isolated from plasma using the AVENIO cfDNA Isolation Kit (Roche, Basel Switzerland). The quality of the DNA was analyzed using QubitTM dsDNA HS assay kit (Thermo Fischer Scientific, Waltham, MA, 125 USA) and 2100 Bioanalyzer (Agilent, Santa Clara, CA, USA). The ctDNA was prepared for sequencing using the AVENIO ctDNA Surveillance Kit (Roche) and sequenced using the 0.198 Mb AVENIO Surveillance panel (Roche), which contains 197 lung cancer-related genes [27]. The ctDNA was sequenced in a multiplex of sixteen samples on a NextSeq 500 High Output Lane (Illumina, San Diego, CA, USA).”

Thank you for answering my question. However, I am still uncertain if any new NGS data were presented in the paper. After the author's publication [24] reading, I assume that all NGS reactions were described in the cited paper and the results shown in this manuscript were previously published. If my assumption is correct, the NGS method should not be described in this manuscript, and any NGS results should be properly cited. However, If authors decide to leave the NGS description in the manuscript a brief description of the most important stages and the amount of DNA used for sequencing individual samples should be included in this work. Additionally, the authors provide a link to the manufacturer's website, which may not be available in the future. Instead, the authors could include additional information related to the methodology in a supplement to ensure it is available to readers in the future.

Respond to Authors Reply 7: “Furthermore, we have clarified the definition of ctDNA negative and positive samples.

Line 153 in revised manuscript: „The ctDNA sequencing data was analyzed using the AVENIO Oncology Analysis 153 Software v. 2.0.0 (Roche). A sample was considered ctDNA positive if any lung cancer related mutations could be detected in the sequencing data following the filtering de-155 scribed in our previously published study [24]. If no mutations were detected, the sample was defined as ctDNA negative.”

Line 350 in revised manuscript: “In a previously published study with the same patient cohort (n=41), excluding one patient due to lack of plasma, we demonstrated that the presence of ctDNA after one or two cycles of treatment (Tx) was associated with shorter PFS and OS [24].

Line 468 in revised manuscript: “No association was found between the absence of ctDNA 468 after treatment initiation and having a high expression of FASLG and ICOSLG at baseline.”

The phrase "ctDNA positive or negative" actually refers to whether or not ctDNA was detected in the plasma and in what amount. Since NGS libraries were constructed, it means that ctDNA was present in all plasma samples. Currently, the authors have written that cfDNA "positive" refers to patients with mutations, while "negative" refers to those without mutations. However, I believe that this is an oversimplification that risks leading to incorrect interpretations of the data. Therefore, it is important to clarify that the terms. I would suggest using the term "ctDNA mutation-positive or negative" instead of "ctDNA positive and negative”. "ctDNA mutation-positive or negative" is more precise than "ctDNA positive and negative." Please clarify it.

3)      Results, 3.2 paragraph “Expression of plasma proteins”, line 192:” Seven patients (16.67%) 192 had KRAS mutations detected in a tissue biopsy before treatment initiation, and two pa-193 tients (4.76%) had a BRAF mutation”

To avoid misleading readers, it's important to differentiate between analyses carried out in previous work and those in the current one. To achieve this, please add the expression " In our previous work [24] we have showed that seven patients (16.67%) 192 had KRAS mutations detected in a tissue biopsy before treatment initiation, and two patients (4.76%) had a BRAF mutation." at the beginning of the sentence if the analyses were part of the previous work. I am not sure here because, for example, the HR values given in the analyses carried out in the previous work (Table S1) do not match with those presented in the current one. This clarification will help readers better understand the context of these analyses.

4)      Figure 3, line237: “Heatmap showing scaled protein expression for the patients at all three time”. Due to the large amount of data and time points analyzed together, it is difficult to discern the conclusion presented in the text: line 233: “Furthermore, the analysis shows that ICOSLG and FASLG cluster together in the heatmap suggesting a similar expression pat-234 tern of the two proteins”. Can you elucidate it?"

Authors Reply 11: We agree with the reviewer that Figure 3 contains a large amount of data that can potentially be overwhelming. We have further elaborated on the findings of the heatmap in the text and have chosen to move the figure to the supplementary files for clarity. Changes in the text: Elaboration of the findings in the heatmap at page 8, line 284 to page 9, line 296. Figure 3 has been moved to Supplementary Figure S3. The following figures and supplementary figures have, as a consequence of this, changed names.

Thank you very much for taking into account the comments and correcting the text. However, moving the figure to the supplement alone does not fully address the issue. The authors state that "Furthermore, the analysis shows a close clustering of ICOSLG and FASLG. The proximity of the two proteins in the heatmap suggests a similar expression pattern of the two proteins." but this is not clearly reflected in the figure provided. Would it be possible to add for example a rectangle or some other visual marker to indicate the locations of both genes? This would greatly enhance the legibility of the figure.

Author Response

We are grateful that the reviewer has taken the time to further review this manuscript giving us the chance to further clarify the methods used. Please find the detailed responses below and the corresponding revisions highlighted in green in the re-submitted files along with the changes from the first round of reviews marked in red.

Major comments:

Comment 1: “The aim of the paper can be misleading. The authors indicate that the main aim of the study was to investigate the immune-related plasma proteins and ctDNA level in order to identify predictive biomarkers that can distinguish patients that respond to pembrolizumab from non-responders (Introduction, line 71). However, in this study all the analysis are based on PFS and OS. There is luck of information about the method of patients categorization as responders or non-responders (how many patients had a partial response (PR), complete response (CR), progressive disease (PD) or stable disease (SD) and appropriate statistical analysis. I would like to ask the authors for a comment on this issue.”

Authors Reply 1: This is a valid point, and we have conducted a Fisher’s exact test to show the association between the high expression FASLG and ICOSLG group and radiologically determined response. Furthermore, we have added information about best overall response to Supplementary Table S1, showing baseline characteristics. We find no association between high FASLG and ICOSLG expression and radiological response, which we have commented on in our revised Discussion section.

We acknowledge that the term ‘responders’ can be misleading, and we have rephrased our aim from identifying responders to predicting clinical outcome.

Changes in the text: We added the definition of best overall response to page 3, line 141-143. Rephrasing of aim at page 2, line 73-74, and the use of the term responders at page 1, line 34-35 and page 13, line 474-475, as well as adding best overall response values to Supplementary Table S1. We added a section to the Discussion regarding radiologically assessed response to immunotherapy at page 12, line 416-430 along with the references 29-34.

Thank you for making the corrections. However, I would suggest also updating the manuscript title.

Reply 1 (second round of review): Thank you for the suggestion. We have updated the title of the manuscript to “Plasma immune proteins and circulating tumor DNA predict clinical outcome in non-small cell lung cancer treated with an immune checkpoint inhibitor”

Changes in the text: Rephrasing of title at page 1, line 2-4.

Comment 2: “Material and Methods, 2.3 paragraph “Olink Proximity Extension Assay”, line 95: there is luck of even shortcut of protein level analysis method (Olink Target 96 Immuno-Oncology panel), which is the core of the paper. The reference [25] (line 100): Olink. Olink Target 96 Immuno-Oncology. Available online: https://www.olink.com/products-services/target/immuno-oncology/ (line 512)“ can be change or delated. Additionally, please indicate briefly what is the NPX and how it is calculated. Using the phrases “protein expression” in the text without clarification may be misleading for some readers.”

Authors Reply 3: We thank the reviewer for the suggestion and have further explained the Olink proximity extension assay and elaborated on the definition of the normalized protein expression (NPX) value. Changes in the text: Explanation of the Olink PEA method and calculation of NPX values at page 3, line 105-122.

I would like to express my gratitude to the authors for providing an extended description of the method. However, the “Material and Methods” chapter is a crucial part of a scientific work. Its purpose is to provide a detailed description of the experiments conducted so that other scientists can replicate and validate the results. I understand that the company "BioXpedia, Aarhus, Denmark" was responsible for protein analysis. However, authors must provide some basic information, for example about the amounts of input material that were analyzed. Without such information, it will be impossible for anyone to verify the results. It is worth noting that references [25] and [26] only provide a link to the manufacturer's website, which contains a brief description of the used Olink Target 96 Immuno-Oncology panel (Olink Prote-110 omics). However, it doesn't include a detailed description of the procedure used by the authors for their samples, including the amounts of reagents used, reaction conditions, etc. Typically, if the reactions are carried out following the protocol recommendation provided by the manufacturer, such information is included in the text, along with a brief description of the procedure. The goal is to give the reader a clear idea of the subsequent steps that were taken. In my opinion, instead of providing a link to an external website that may disappear in the future, the authors should include additional information related to the methodology in a supplement.

Reply 2 (second round of review): 1 µL of plasma was used for each patient. BioXpedia analyzed the samples following the manufacturers protocol. We agree that external websites may only exist temporarily and have removed the references to websites to avoid confusion.

Changes in the text: We have added information stating that BioXpedia analyzed the samples following the manufacturers protocol (page 3, line 101-102). Furthermore, we have provided information about the volume of plasma used and the controls used on each 96-well plate (page 3, line 110-113). Reference 25 and 26 have been removed.

Minor comments:

Comment 3: Material and Methods, 2.1 paragraph “Patients”, line 77: corrected sentence: “Pathological detection of mutations in tissue biopsies was done as part of the standard procedure for lung cancer patients 78 at Aarhus University Hospital.”

Was this sentence discussing the detection of pathogenic mutations? I'm not sure of the author's intent. Please clarify it.

Reply 3 (second round of review): We have included this information due to a wish from Reviewer 1, regarding information about pathogenic variants identified in the tissue biopsy. This was not part of the study, but pathological detection of mutations in tissue biopsies is always done as part of the standard procedure for lung cancer patients in Denmark, and thus, this information was available to us. Looking into the mutations detected in the tissue, we found that seven patients had a KRAS mutation and two patients had a BRAF mutation in a tissue biopsy. We found no association between the KRAS mutation in a tissue biopsy and survival, which we have described at page 4, line 187-191.

Changes in the text: None.

Comment 4:  Material and Methods, 2.4 paragraph “Next-Generation Sequencing”, line 103: The Next-Generation Sequencing method used in this paper is not related to the results presented. To make the paper more informative, I suggest adding a description of the isolation and measurement method for ctDNA at this point. Furthermore, it would be highly beneficial to include a table showing the correlation between ctDNA levels and protein levels.

Authors Reply 7: We have added a description of the isolation and library preparation to the Next-Generation Sequencing data. Furthermore, we have clarified the definition of ctDNA negative and positive samples. Changes in the text: We have added a longer description of the sequencing method at page 3, line 126-132. The ctDNA analysis software and definition of ctDNA positive and negative samples have been added at page 4, line 156-160

Line 122 in revised manuscript: “Sample preparation and sequence analysis of ctDNA was performed as part of a previously published study [24]. Briefly, cell-free DNA was isolated from plasma using the AVENIO cfDNA Isolation Kit (Roche, Basel Switzerland). The quality of the DNA was analyzed using QubitTM dsDNA HS assay kit (Thermo Fischer Scientific, Waltham, MA, 125 USA) and 2100 Bioanalyzer (Agilent, Santa Clara, CA, USA). The ctDNA was prepared for sequencing using the AVENIO ctDNA Surveillance Kit (Roche) and sequenced using the 0.198 Mb AVENIO Surveillance panel (Roche), which contains 197 lung cancer-related genes [27]. The ctDNA was sequenced in a multiplex of sixteen samples on a NextSeq 500 High Output Lane (Illumina, San Diego, CA, USA).”

Thank you for answering my question. However, I am still uncertain if any new NGS data were presented in the paper. After the author's publication [24] reading, I assume that all NGS reactions were described in the cited paper and the results shown in this manuscript were previously published. If my assumption is correct, the NGS method should not be described in this manuscript, and any NGS results should be properly cited. However, If authors decide to leave the NGS description in the manuscript a brief description of the most important stages and the amount of DNA used for sequencing individual samples should be included in this work. Additionally, the authors provide a link to the manufacturer's website, which may not be available in the future. Instead, the authors could include additional information related to the methodology in a supplement to ensure it is available to readers in the future.

Reply 4 (second round of review): The reviewer is correct that the sequencing of ctDNA was done as part of a former study (reference 24), and a description of the method can be found in that paper. We have therefore chosen to follow the reviewer’s suggestion and have removed the description of the method and kept the reference to the previously published study. We agree that this will improve the transparency of the origin of the NGS data.

Changes in the text: Removal of description of NGS analysis, and instead referring to the formerly published paper (reference 24) at page 3, line 124-125.

Comment 5: Respond to Authors Reply 7: “Furthermore, we have clarified the definition of ctDNA negative and positive samples.

Line 153 in revised manuscript: „The ctDNA sequencing data was analyzed using the AVENIO Oncology Analysis 153 Software v. 2.0.0 (Roche). A sample was considered ctDNA positive if any lung cancer related mutations could be detected in the sequencing data following the filtering de-155 scribed in our previously published study [24]. If no mutations were detected, the sample was defined as ctDNA negative.”

Line 350 in revised manuscript: “In a previously published study with the same patient cohort (n=41), excluding one patient due to lack of plasma, we demonstrated that the presence of ctDNA after one or two cycles of treatment (Tx) was associated with shorter PFS and OS [24].

Line 468 in revised manuscript: “No association was found between the absence of ctDNA 468 after treatment initiation and having a high expression of FASLG and ICOSLG at baseline.”

The phrase "ctDNA positive or negative" actually refers to whether or not ctDNA was detected in the plasma and in what amount. Since NGS libraries were constructed, it means that ctDNA was present in all plasma samples. Currently, the authors have written that cfDNA "positive" refers to patients with mutations, while "negative" refers to those without mutations. However, I believe that this is an oversimplification that risks leading to incorrect interpretations of the data. Therefore, it is important to clarify that the terms. I would suggest using the term "ctDNA mutation-positive or negative" instead of "ctDNA positive and negative”. "ctDNA mutation-positive or negative" is more precise than "ctDNA positive and negative." Please clarify it.

Reply 5 (second round of review): We agree that the terms ctDNA positive and negative could be misunderstood, and the terms ctDNA mutation positive and negative could help avoid misinterpretations. We have revised our terminology according to the reviewer’s recommendation.

Changes in the text: ctDNA terminology changed throughout the manuscript.

Comment 6: Results, 3.2 paragraph “Expression of plasma proteins”, line 192:” Seven patients (16.67%) 192 had KRAS mutations detected in a tissue biopsy before treatment initiation, and two pa-193 tients (4.76%) had a BRAF mutation”

To avoid misleading readers, it's important to differentiate between analyses carried out in previous work and those in the current one. To achieve this, please add the expression " In our previous work [24] we have showed that seven patients (16.67%) 192 had KRAS mutations detected in a tissue biopsy before treatment initiation, and two patients (4.76%) had a BRAF mutation." at the beginning of the sentence if the analyses were part of the previous work. I am not sure here because, for example, the HR values given in the analyses carried out in the previous work (Table S1) do not match with those presented in the current one. This clarification will help readers better understand the context of these analyses.

Reply 6 (second round of review): The data mentioned here refer to the pathologically detected mutations in tissue biopsies and are not results from our previous study. These data were added in response to a request from Reviewer 1, as mentioned in Reply 3. In our previous work, only data from ctDNA was used and no tissue mutations was included. The data mentioned here have therefore never been published before, and the reference to our previous work will not be applicable.

Changes in the text: None.

Comment 7: Figure 3, line237: “Heatmap showing scaled protein expression for the patients at all three time”. Due to the large amount of data and time points analyzed together, it is difficult to discern the conclusion presented in the text: line 233: “Furthermore, the analysis shows that ICOSLG and FASLG cluster together in the heatmap suggesting a similar expression pat-234 tern of the two proteins”. Can you elucidate it?"

Authors Reply 11: We agree with the reviewer that Figure 3 contains a large amount of data that can potentially be overwhelming. We have further elaborated on the findings of the heatmap in the text and have chosen to move the figure to the supplementary files for clarity. Changes in the text: Elaboration of the findings in the heatmap at page 8, line 284 to page 9, line 296. Figure 3 has been moved to Supplementary Figure S3. The following figures and supplementary figures have, as a consequence of this, changed names.

Thank you very much for taking into account the comments and correcting the text. However, moving the figure to the supplement alone does not fully address the issue. The authors state that "Furthermore, the analysis shows a close clustering of ICOSLG and FASLG. The proximity of the two proteins in the heatmap suggests a similar expression pattern of the two proteins." but this is not clearly reflected in the figure provided. Would it be possible to add for example a rectangle or some other visual marker to indicate the locations of both genes? This would greatly enhance the legibility of the figure.

Reply 7 (second round of review): We have added arrows pointing at FASLG and ICOSLG along with boxes in the heatmap in Supplementary Figure S3, to enhance the understanding of the figure.

Changes in the text: Added arrows and boxes to FASLG and ICOSLG in Supplementary Figure S3.

Round 3

Reviewer 3 Report

Comments and Suggestions for Authors

The authors have adequately addressed all the comments, from the previous review. I have only one minor suggestion to improve the article regarding comment 3.

Comment 3: Material and Methods, 2.1 paragraph “Patients”, line 78: corrected sentence: “Pathological detection of mutations in tissue biopsies was done as part of the standard procedure for lung cancer patients 78 at Aarhus University Hospital.”

Was this sentence discussing the detection of pathogenic mutations? I'm not sure of the author's intent. Please clarify it.

Reply 3 (second round of review): We have included this information due to a wish from Reviewer 1, regarding information about pathogenic variants identified in the tissue biopsy. This was not part of the study, but pathological detection of mutations in tissue biopsies is always done as part of the standard procedure for lung cancer patients in Denmark, and thus, this information was available to us. Looking into the mutations detected in the tissue, we found that seven patients had a KRAS mutation and two patients had a BRAF mutation in a tissue biopsy. We found no association between the KRAS mutation in a tissue biopsy and survival, which we have described at page 4, line 187-191.

Changes in the text: None.

Thank You very much for your explanation. To improve clarity, I would suggest revising the sentence as follows:: “The detection of mutations in tissue biopsies was done by a pathologist, as part of the standard procedure for lung cancer patients at Aarhus University Hospital.”